# Peer review of "PD-L1 Inhibitors: Different Classes, Activities, and Mechanisms of Action"

_ijms, 2021, doi:10.3390/ijms222111797_

Round 1

Reviewer 1 Report

In this manuscript, Surmiak E et al. compared the PD-1/PD-L1 dissociation capacities and in vitro bioactivities of the different types of PD-L1 inhibitors. Based on the findings of the study, the authors proposed that stringent in vitro cell-based assays should be coupled with structure-based PD-L1/PD-1 dissociation studies, and determination of species specificity (for any off-target effects) of the inhibitor under test prior to evaluation of their potency/efficacy in the mouse models should be performed. The authors conclude that these two steps would be important in the search for newer therapeutic inhibitors that will show efficacy close to therapeutic antibodies. 

Overall, the study is interesting and has clinical significance. However, the manuscript needs to be revised before gets accepted for publication. Below mentioned the points that needed clarification.   

Major specific comments.

  1. Line 88-90. The authors should describe the specific cytotoxic activities seen when a higher concentration of the small molecules was used for the testing. If not tested, authors can cite previous reports showing these adverse effects.
  2. Figure 3. Inhibitor-PD-L1 binding assay. Structure-based data: Authors should provide how structures of the inhibitors-PD-L1 complex are derived. What parameters were used?   
  3. PD-L1 dimer formation. The behaviour of the protein in the solution is completely different from the membrane-embedded PD-L1 found in the cancer cells. Is there any biological evidence of PD-L1 dimer formation?  
  4. PD-L1 maturation and trafficking. What cell types were used in other papers to conclude that some inhibitors induce defects in the PD-L1 maturation and trafficking? Does these cells had forced/transgenic expression of PD-L1? These issues need to be discussed thoroughly, as the cell types, cancer versus cell lines; and type of PD-L1 expression, transgenic versus intrinsic expression in cancer phenotype, may represent confounding factors. This can influence the conclusion of the findings.     
  5. What positive and negative controls were used for the HTRF assay. 
  6. HTRF assay. Why selectively 1:10 molar ratio of hPD-L1 and hPD-1 used? 
  7. How HTRF assay performed. What fluorochrome donor and acceptor were used? The methods of HTRF assay need to be described in detail.  
  8. ICB assay. What is the source of luminescence measured in the assay? How does it relate to the activation of the Jurkat cells? The authors should provide detailed methods.  
  9. ICB assay. It appears that CHO/TCRAct/PD-L1 cells mimic the antigen-presenting cells that present TCR and inhibitory PD-L1 signal to the PD-1 expressing Jurkat T cells. The limitation of the assay is, CHO cells are not cancerous and therefore lack cancer-associated phenotype and signals that they provide to T cells in the culture. The best way to evaluate the EC50 of the compound is to use cancer cells that naturally express PD-L1. Also, state the specificity of the PD-L1 molecules transgenically expressed in the CHO cells. 
  10. Also, Jurkat T cells used are of CD4+ in nature. All the activity measured by using Jurkat will reflect the response of CD4 T cells. Although rejuvenation of exhausted CD4’s in tumors is important, it is CD8+ T cells, owing to their direct cytotoxic properties, that contribute significantly in getting rid of the tumor. The authors should briefly highlight this limitation of the assay. 
  11. The recent development of the 3D cell cultures, organoids systems (derived from cancer patients) helping to improve the model system to test the efficacy of the ICB inhibitors (including anti-PD-L1/PD-1) against the variety of cancer types (Boucherit N et al., Front Immunol, 2020; Scognamiglio, G et al., Br J Cancer, 2019; Mikaela Grönholm et al., Cancer Res, 2021; Liu L et al., J Transl Med, 2021). This would reduce the burden on heavily reliant on mouse/rodent model system, which does not mimic true human cancer microenvironment. The authors should comment on the pros and cons of these novels models to test the effectiveness of the ICB inhibitors.     

Minor comments.

- Mention in the text the full form of ICMs.

-There are punctuation errors at several places in the text. 

Reviewer 2 Report

Please add discussion on biological relevance such as implications in using cancer-bearing murine models. Extend the discussion on potential tumor types that the current study can benefit. Please extend the description of HTRF method for general readers.
